# Test on Compressive Performance of Concrete Filled Circular Steel Tube Connected by Thread through Inner Lining Tube

**DOI:** 10.3390/ma15238619

**Published:** 2022-12-02

**Authors:** Qingli Wang, Yijing Zhang, Kuan Peng

**Affiliations:** 1School of Civil Engineering, University of Science and Technology Liaoning, Anshan 114044, China; 2School of Mechatronic Engineering, Southwest Petroleum University, Chengdu 610500, China

**Keywords:** circular concrete-filled steel tube, inner lining tube, threaded connection, axial compressive performance

## Abstract

The connection method of lengthening the steel tube of concrete filled circular steel tubes by inner lining tube and threaded connection is proposed. Taking the length, depth, and position of the thread as the basic parameters, 12 concrete filled circular steel tubes connected by thread through inner lining tube were designed and manufactured, and an axial compressive test was carried out. The axial compressive loading-longitudinal compressive displacement curves, axial compressive loading-strain of steel tube curves, and failure mode of the specimens were analyzed, and the effects of different parameters on the axial compressive bearing capacity and stiffness of the specimens were studied. The results show that the axial compressive loading-longitudinal compressive displacement curves of the specimen can be divided into the elastic stage, elasto-plastic stage, and plastic reinforcement stage in the range of parameters studied in this paper. The bearing capacity and stiffness of the specimens connected by thread through inner lining tube are no worse than those of the unconnected specimen or the specimen connected by weld. Bearing capacity and stiffness of the specimen increase with the increase of thread length. The calculation method of the axial compressive bearing capacity of concrete filled circular steel tubes connected by thread through inner lining tube are suggested.

## 1. Introduction

Concrete filled steel tube structure has the advantages of high bearing capacity and a good economic effect. In recent years, it has been applied more often and studied in the field of construction engineering [1,2,3,4]. Due to the limitations of raw material length, hoisting, and transportation capacity, concrete filled steel tubes often have the problem of lengthening steel tubes in the construction process. [5]. Welding (Figure 1a) is one of the main ways to lengthen the steel tube of concrete-filled steel tubes at present. It has the advantages of not being affected by the size of the specimen and a convenient construction process. The stress concentration is easy to occur at the weld line [6], and the welding quality is easily affected by anthropic, climate, environment, and other factors, resulting in some quality problems [7]. In addition, there are flange connections (Figure 1b) and grouting sleeve connections (Figure 1c) for the extension of concrete-filled steel tubes.

Luciano et al. [8] conducted an experimental study on the bending and bearing capacity of concrete-filled steel tubes connected by inner and outer double-layer flanges, and revealed the distribution of their internal forces by analyzing the strains of steel tubes and bolts. Chen Y et al. [9] analyzed the bending performance of internal and external rigid flanges through tests and the finite element method. Guo et al. [10] systematically studied the interface nonlinearity and force transmission of bolted flange joints under impact load. There are still some problems in flange connections in practical engineering, such as changing the shape of the outer surface of the steel tube at the joint connection and not being conducive to the loading transfer at the joint. Chen [11] proposed a new method of concrete-filled steel tube connection with a grouting sleeve. Starting with the analysis of design and mechanical principle, the design parameters, detailed structure, and scope of application of this new connection are introduced in detail. Wu [12] carried out experimental research on the flexural performance of concrete-filled steel tubular specimens and comparison specimens connected by grouting sleeves, and analyzed their stiffness, ductility, and stress process. For the connection of the grouting sleeve, the inability to accurately control the grouting amount [13] is likely to lead to the lack of grout at the connection of components, or the increase of the outer contour [14], which will lead to quality problems in the structure [15]. In view of this, it is necessary to provide a new connection mode with a high degree of standardization, convenient construction, reasonable mechanical properties, and not susceptible to environmental and anthropic factors, and further realize the diversity of concrete filled steel tube extension.

The thread uses the simple mechanical incline principle, and it is used reasonably. In the production process, standardized pipelining can be adopted. It is convenient to assemble in the construction process, which greatly saves the construction period and fully ensures the structural quality. At present, a threaded connection has been widely used in machining, industrial production, and assembly manufacturing, and it reached technological maturity. Therefore, this paper attempts to apply a threaded connection to the extension of steel tubes in concrete-filled steel tube structures. If the steel tube is directly connected with internal and external threads, the threaded connection will be weakened by at least 50%, thereby reducing the bearing capacity. In order to avoid this situation, the inner lining tube was designed as an excessive connection tube, as shown in Figure 2.

Magnesite is a magnesium carbonate mineral (Mg CO_3_), which is also the main source of magnesium resources [16]. In areas rich in magnesite resources, high-grade magnesium ore was preferentially selected in the process of development and utilization, the resulting large numbers of low-grade magnesite and magnesite tail ore are idle, causing a great waste of resources and serious environmental pollution. How to deal with and utilize low-grade magnesite is the general problem to be solved at present [17]. This paper attempts to use low-grade magnesite mining as the coarse aggregate of concrete to make low-grade magnesite concrete filled circular steel tubes. Overall, the research idea of low-grade magnesite concrete filled circular steel tubes connected by thread through inner lining tube is proposed.

For the threaded concrete-filled steel tube, axial tension and torsion are the most unfavorable stress forms, followed by tension-bending, bending, and compression-bending. Axial compression and shear are favorable stress forms for threaded concrete-filled steel tube, and the axial compressive performance is the most important and the most basic performance of concrete-filled steel tubes. Therefore, the axial compressive performance of threaded concrete-filled steel tubes should be studied firstly.

This paper takes the axially compressed stub column of low-grade magnesite concrete filled circular steel tubes connected by thread through inner lining tube as the research object, and carries out experiments to analyze the axial compressive loading (*N*)-longitudinal compressive displacement (∆) curve, axial compressive loading (*N*)—strain of steel tube (*ε*_s_) curve, bearing capacity, stiffness, and failure mode of the specimen. According to the calculation expression of the bearing capacity proposed in different references, the calculated results are compared with the test results, and suggest the calculation method of axial compressive bearing capacity of low-grade magnesite concrete filled circular steel tubes connected by thread through inner lining tube, and provide some experimental references for the establishment of relevant finite element models in the future [18,19].

## 2. Test Survey

### 2.1. Specimen Design

A total of 12 low-grade magnesite concrete filled circular steel tubes connected by thread through inner lining tube specimens, 2 welded low-grade magnesite concrete filled circular steel tube specimens, and 1 ordinary low-grade magnesite concrete filled circular steel tube specimen were designed. The main parameters of the threaded connection specimens include the length, depth, and position of thread, the outer diameter of the steel tube *D*_s_ = 133 mm, the length of the specimen *L* = 3*D*_s_, the wall thickness of the steel tube *t*_s_ = 6 mm, and the wall thickness of the inner lining tube *t*_is_ = 8 mm. The thread length *l* is calculated as *D*_s_/8, *D*_s_/4 and *D*_s_/2, respectively [20,21], the thread depth *h* is taken as 0.1 *t*_s_ and 0.15 *t*_s_ respectively [22,23], and the thread position is taken as the middle section and end section. Parameters of all specimens are shown in Table 1.

The steel tube confinement effect coefficient *ξ* [1] of the specimens is 2.29, Where:*ξ* = (*A*_s_*f*_y_)/(*A*_c_*f*_ck_) (1)
where: *A*_s_ is the cross-sectional area of steel tube, *f*_y_ is the yield strength of steel tube, *A*_c_ is the cross-sectional area of concrete, *f*_ck_ = 0.67*f*_cu_ is the characteristic axial compressive strength of concrete, *f*_cu_ is the cubic compressive strength of concrete.

### 2.2. Material Properties

#### 2.2.1. Steel

The indices of steel tubes measured according to China National Standard System: *metallic materials- tensile testing- part 1: Method of at room temperature* are shown in Table 2, Where *f*_u_ is the tensile strength of steel tube, *E*_s_ is the elastic modulus of steel tube, vs. is the Poisson’s ratio of steel tube, and *δ* is the elongation of steel tube.

#### 2.2.2. Concrete

Table 3 is mixed proportion of concrete. The main material used are: Portland cement with strength grade 525; running water; sand; particle size of low-grade magnesite is 5~15 mm.

Compressive strength of concrete is *f*_cu_ = 57 MPa, and elastic modulus of concrete *E*_c_ = 31.5 GPa through test.

### 2.3. Specimen Fabrication

The schematic diagram of steel components of some specimens is shown in Figure 3, in which the unit of dimensions is mm.

All specimens after pouring concrete are shown in Figure 4.

### 2.4. Loading and Measurement

Test equipment is shown in Figure 5. Four displacement meters are arranged at intervals of 90° around the specimen to measure the overall longitudinal deformation of the specimen. One transverse and one longitudinal strain gauge are pasted on the outer wall of the steel tube along the transverse direction of the specimen at interval of 90° to measure the transverse and longitudinal strains of the steel tube.

After adjusting of the specimen and equipment, the specimen is step loaded according to the estimated bearing capacity [1]. Within the elastic range, the loading of each stage is 1/10 of the estimated bearing capacity, the instrument data is recorded after each stage of loading, and the next stage of loading is carried out after holding the load for 2 min until it reaches 60% of the estimated bearing capacity. Then, load at the rate of 2 kN/s until the compressive displacement reaches 30 mm (about 75,000 με).

## 3. Results and Discussion

### 3.1. Specimen Failure Model

Figure 6 shows the failure modes of the ordinary specimen and the welded specimen. Both specimens show outward buckling failure bounded by the end plate/weld line (which was caused by the higher weld strength due to full penetration weld is adopted), which was consistent with the results of reference [1].

The typical failure mode of the threaded connection specimen with short thread (*l* = 16.5 mm) is shown in Figure 7. The eversion deformation at the butt joint of the tube (Figure 2) and the outward buckling at the end of the specimen are mainly failure characteristics. When the steel tube is cut, it can be found that the crushed part of the concrete is mainly located at the butt joint of the tube and the end of the specimen. The deformation of the concrete of all specimens corresponds to the deformation of the steel tube, indicating that the deformation of the steel tube and concrete is consistent, and the concrete has good plastic filling performance.

The typical failure modes of the threaded connection specimens with long thread (*l* = 33 mm, 66 mm) are shown in Figure 8. Outward buckling at the edge of the inner lining tube (Figure 2) and the end of the specimen are mainly failure characteristics. When the steel tube is cut, it can be found that the crushed part of the concrete is mainly located at the edge of the inner lining tube and the end of the specimen.

### 3.2. Test Curves

#### 3.2.1. Axial Compressive Loading-Longitudinal Compressive Displacement Curves

The axial compressive loading-longitudinal compressive displacement curves of ordinary specimen and welded specimens are shown in Figure 9. It can be seen that the trend of all curves is roughly the same, the stiffness of the welded specimens is basically the same as that of the ordinary specimen, and the bearing capacity is slightly higher than that of the ordinary specimen. This is because in order to ensure the objectivity of the comparative test, full penetration weld and the large weld leg size are adopted in MW specimen and EW specimen.

Figure 10 shows the *N*-∆ curves of threaded connection specimen and welded specimen. It can be seen that the bearing capacity and stiffness of the threaded connection specimen are not inferior to those of the welded specimen.

It can also be seen from Figure 9 to Figure 10 that all specimens are in the linear elastic stage at the initial stage of loading, and the relationship between loading and displacement is linear. As the loading continues to increase, the specimen enters the elasto-plastic stage. Finally, the curves enter the enhanced section due to the large constraint effect coefficient of the steel tube.

#### 3.2.2. Analysis of Loading—Strain Curve

Figure 11 shows the (axial compressive loading) *N*-*ε*_s_ (steel tube strain) curve of all specimens. It can be seen that the steel tube is compressed in a longitudinal direction and tensioned in transverse direction. In the early stage of loading, the longitudinal strain is larger than the transverse strain, and in the late stage of loading, the transverse strain of steel tube increases significantly. It shows that the steel tube has significant lateral restraint on the concrete.

## 4. Analysis of Influencing Factors of Specimens Connected by Thread through Inner Lining Tube

### 4.1. Thread Length

Figure 12 shows the effect of thread length on the *N*-∆ curve of the specimens. It can be seen that the bearing capacity and stiffness of the specimen increase with the increase of the thread length. This is because the greater the increase in the length of the thread, the longer the length of the inner lining tube, and the better the constraint of the threaded section on the concrete.

### 4.2. Thread Position

Figure 13 shows the influence of thread position on the *N*-∆ curve of the specimen. It can be seen that the stiffness and bearing capacity of the specimen connected at the end section are higher than those of the specimen connected at middle section, under other conditions unchanged. When the thread length increases to 33 mm and 66 mm, the bearing capacity significant increases.

### 4.3. Thread Depth

Figure 14 shows the influence of thread depth on the *N*-∆ curve of the specimen. It can be seen that there are few differences between the two threads depth in this test; the change of the thread depth has no significant effect on the axial compressive performance of the specimen under the same other conditions.

## 5. Calculation of Load Bearing Capacity

For axially compressed stub columns, the connection length accounts for a large proportion relative to the overall length of the specimens connected by thread through inner lining tube, but the connection length accounts for a small proportion in the actual engineering. Therefore, the influence of the thickness of inner lining tube can be ignored. In addition, the loading value is defined as *N*_ue_ [1] when the strain reaches [1300 + 12.5*f* ’_c_ + (600 + 33.3*f* ’_c_)*ε*^0.2^]με, *N*_uc_ is the calculated value of bearing capacity, and *f* ’_c_ is the strength of the concrete cylinder. Table 4 shows the comparison of the calculation and test values of the bearing capacity. Average value results show that the result of reference [1] is close to 1, and the mean square deviation result of reference [1] is the largest. It can be seen that the results calculated according to the method provided in reference [1] agree well with this paper.

## 6. Conclusions

Taking the length, depth, and position of the thread as the basic parameters, 12 concrete filled circular steel tubes connected by thread through inner lining tube were designed and manufactured, and an axial compressive test was carried out. The axial compressive loading-longitudinal compressive displacement curves, axial compressive loading-strain of steel tube curves, and failure mode of the specimens were analyzed, and the effects of different parameters on the axial compressive bearing capacity and stiffness of the specimens were studied, and the calculation method of the axial compressive bearing capacity of concrete filled circular steel tubes connected by thread through inner lining tube is suggested for construction industries. The main conclusions can be summarized as follows:

(1) The *N*-∆ curves of all specimens can be divided into elastic stage, elasto-plastic stage and plastic reinforcement stage. All specimens are in the linear elastic stage at the initial stage of loading, and the relationship between loading and displacement is linear. As the loading continues to increase, the specimen enters the elasto-plastic stage. Finally, the curves enter the enhanced section due to the large constraint effect coefficient of the steel tube.

(2) The bearing capacity and stiffness of the specimens connected by thread through inner lining tube are not inferior to the welded specimen or the ordinary specimen. The bearing capacity and stiffness of the specimens connected by thread through inner lining tube increase with the increase of the thread length.

(3) Suggestions for calculating the axial compressive bearing capacity of low-grade magnesite concrete filled circular steel tube connected by thread through inner lining tube are suggested.

## Figures and Tables

**Figure 1 materials-15-08619-f001:**
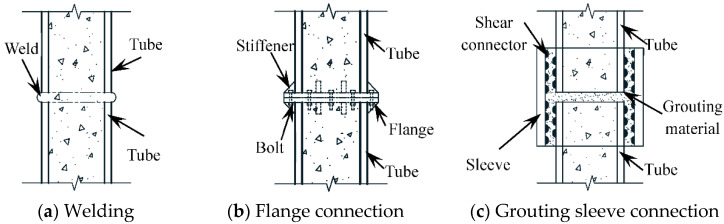
Common lengthening methods of steel tube of concrete filled steel tube.

**Figure 2 materials-15-08619-f002:**
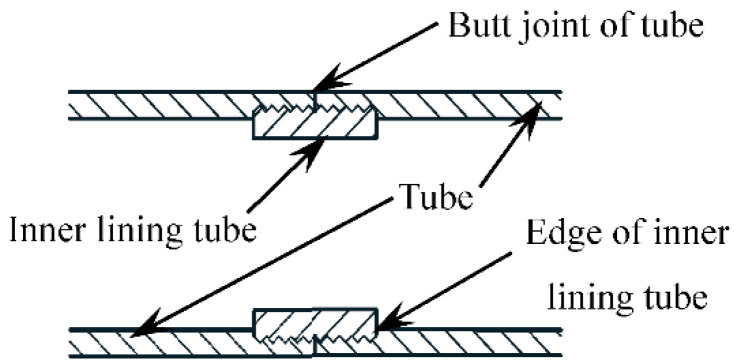
Schematic diagram of steel tube lengthened by thread through inner lining tube.

**Figure 3 materials-15-08619-f003:**
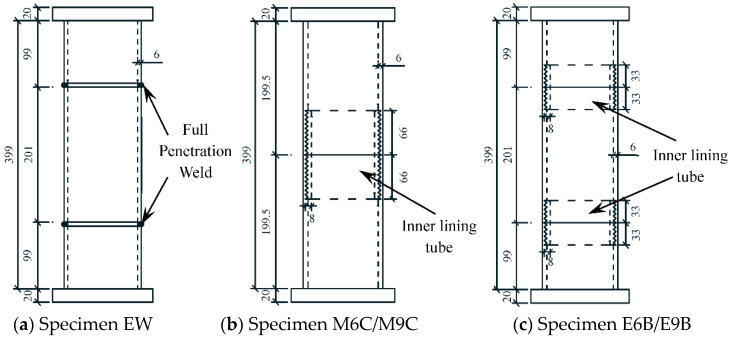
Schematic diagram of steel components of some specimens.

**Figure 4 materials-15-08619-f004:**
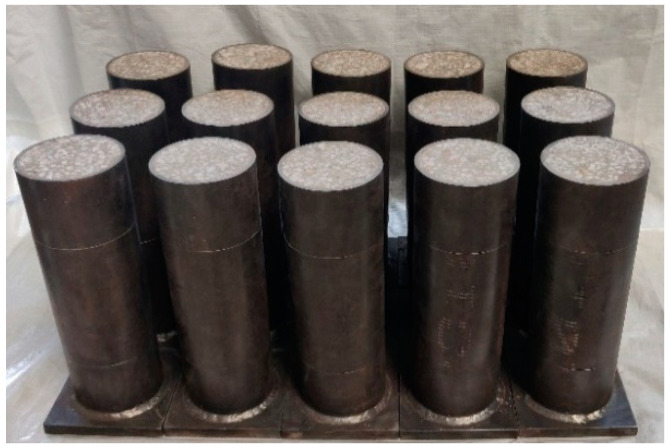
All specimens after pouring concrete.

**Figure 5 materials-15-08619-f005:**
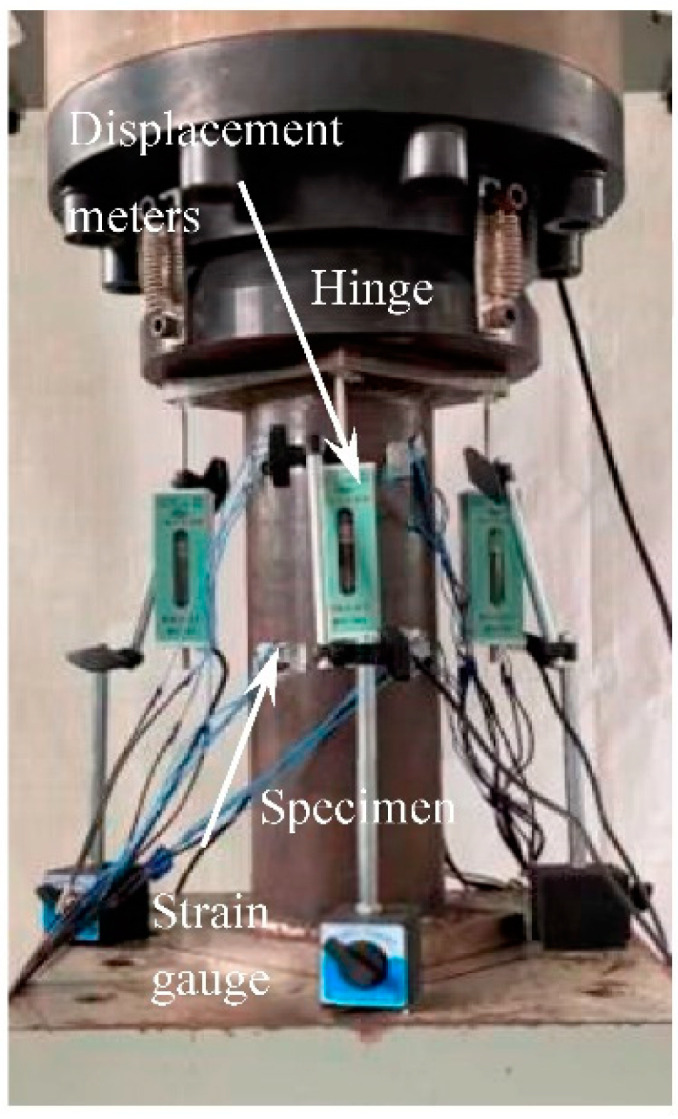
Test equipment.

**Figure 6 materials-15-08619-f006:**
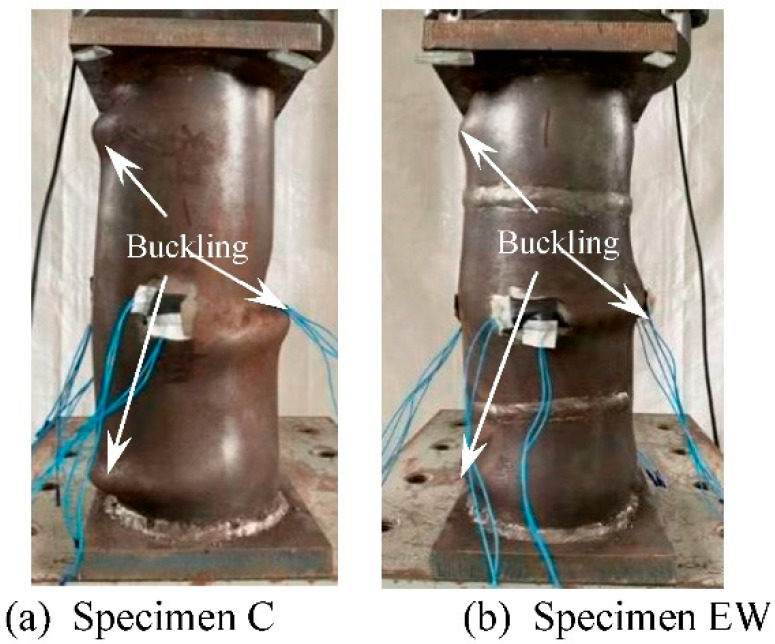
Failure modes of specimens without thread.

**Figure 7 materials-15-08619-f007:**
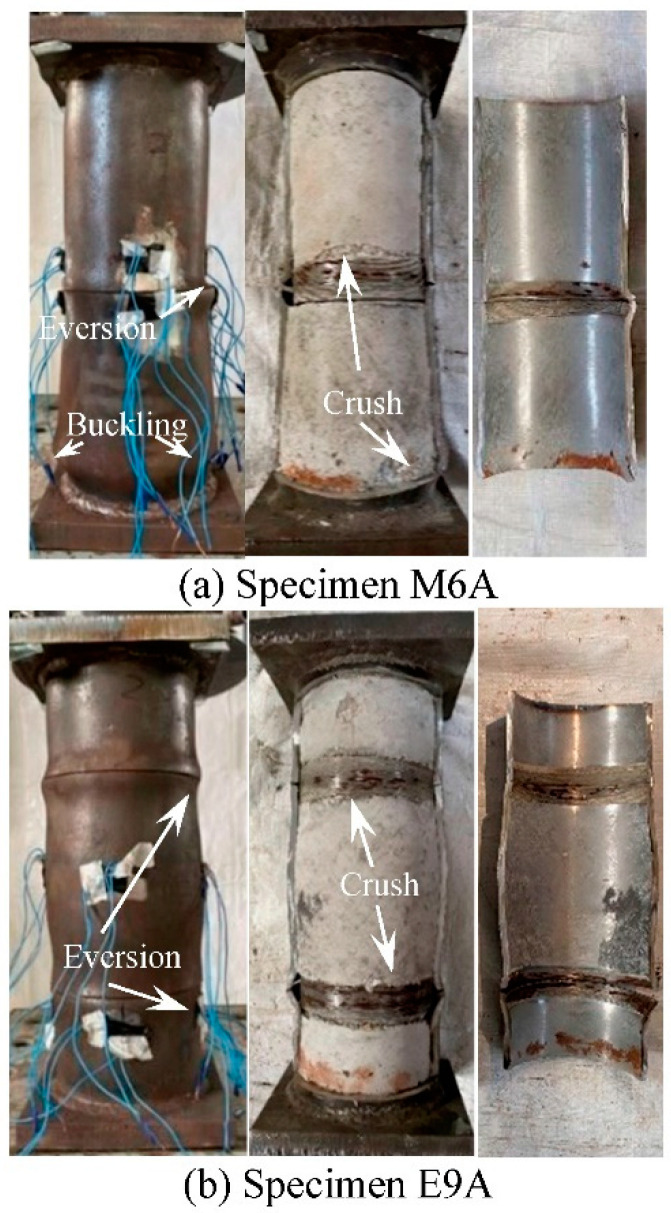
Failure modes of specimens with short thread.

**Figure 8 materials-15-08619-f008:**
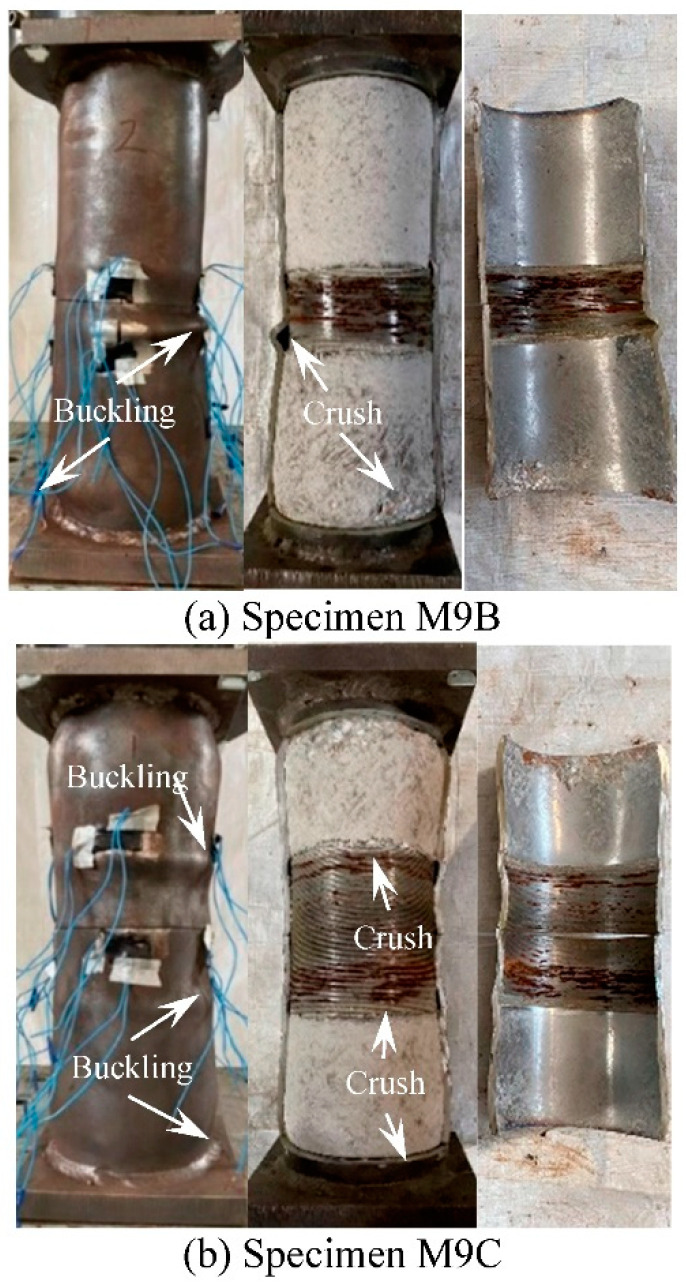
Failure modes of specimens with long thread.

**Figure 9 materials-15-08619-f009:**
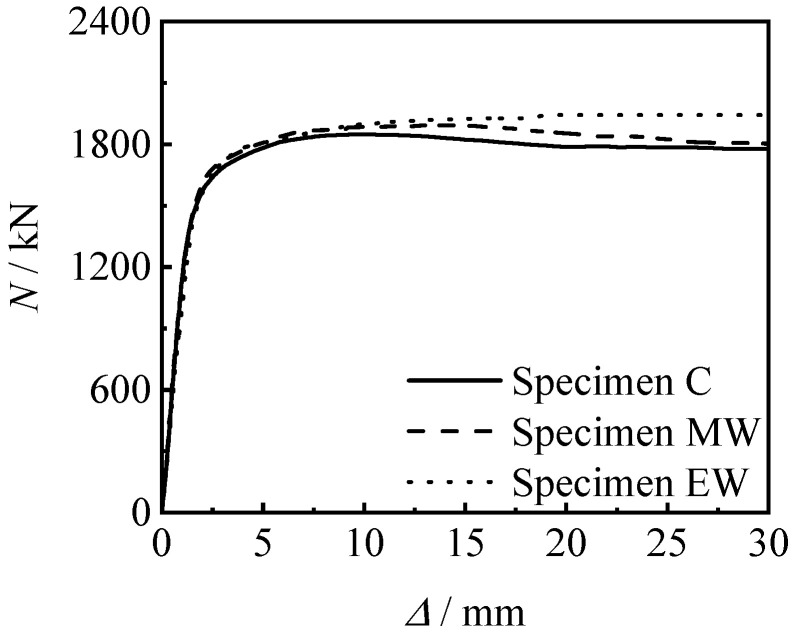
*N*-∆ curves of specimens without thread.

**Figure 10 materials-15-08619-f010:**
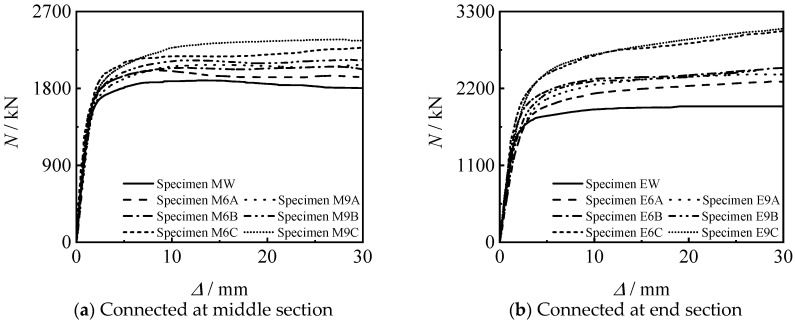
*N-*∆ curves of connected specimens.

**Figure 11 materials-15-08619-f011:**
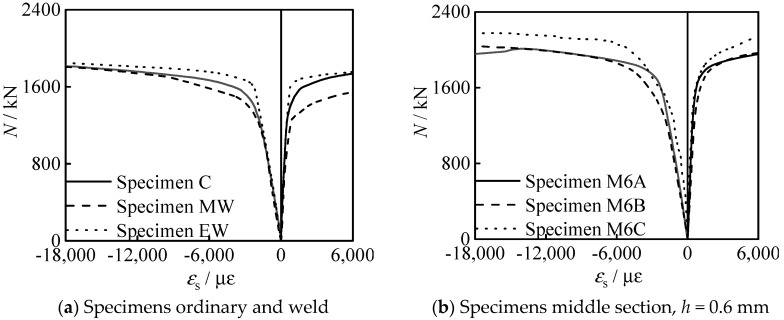
*N-ε*_s_ curves of all specimens.

**Figure 12 materials-15-08619-f012:**
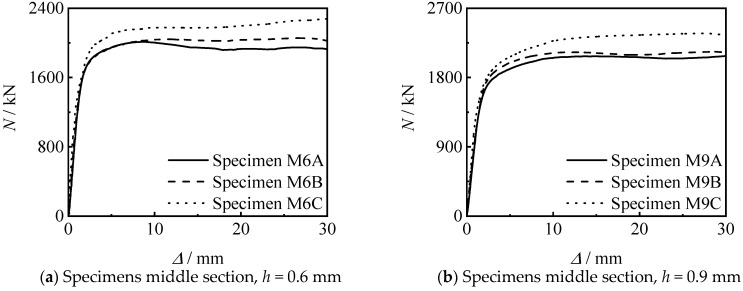
Effect of thread length on *N-*∆ curves of specimens.

**Figure 13 materials-15-08619-f013:**
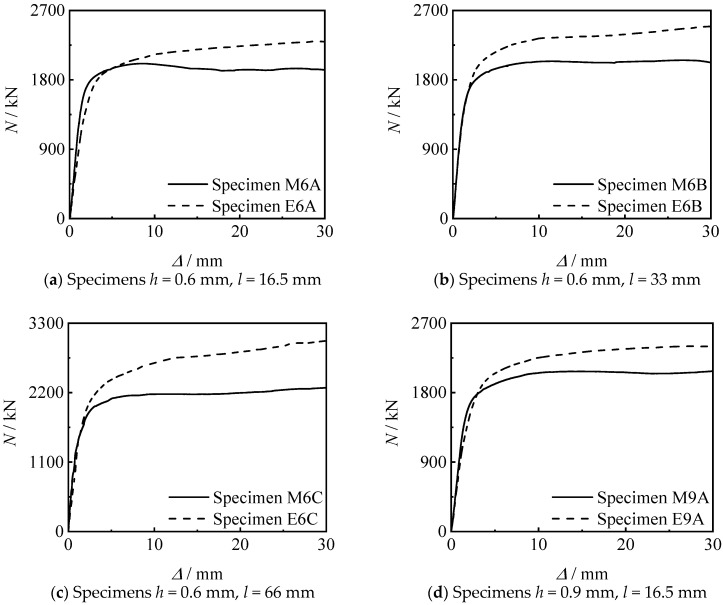
Effect of thread position on *N-*∆ curves of specimens.

**Figure 14 materials-15-08619-f014:**
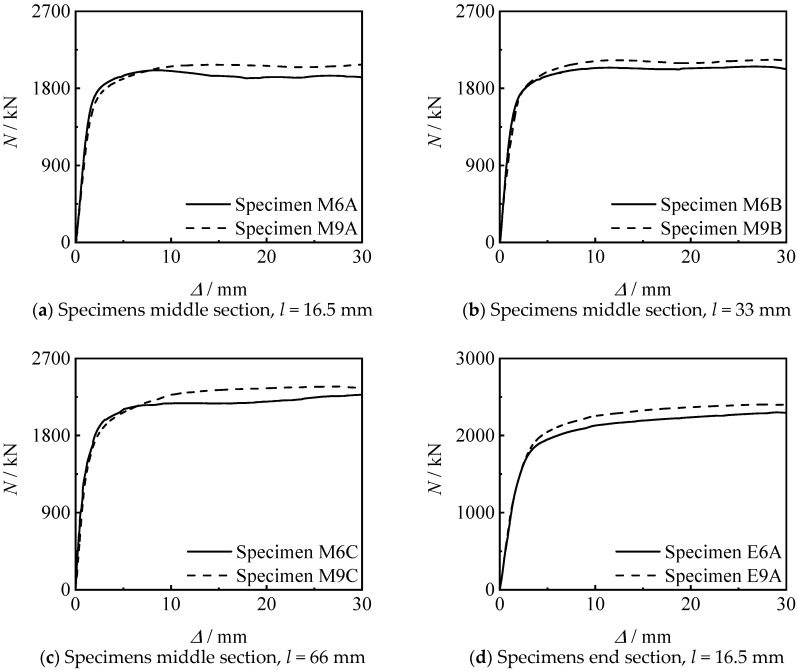
Effect of thread depth on *N-*∆ curves of specimens.

**Table 1 materials-15-08619-t001:** Parameters of specimens.

Serial Number	No.	*D*_s_ × *t*_s_ × *L*/mm	Weld/Thread Position	*h*/mm	*l*/mm
1	C	133 × 5.5 × 399	–	–	–
2	MW		Middle section	–	–
3	EW		End section	–	–
4	M6A		Middle section	0.6	16.5
5	M6B		Middle section	0.6	33
6	M6C		Middle section	0.6	66
7	M9A		Middle section	0.9	16.5
8	M9B		Middle section	0.9	33
9	M9C		Middle section	0.9	66
10	E6A		End section	0.6	16.5
11	E6B		End section	0.6	33
12	E6C		End section	0.6	66
13	E9A		End section	0.9	16.5
14	E9B		End section	0.9	33
15	E9C		End section	0.9	66

Note: No. C is ordinary specimen; MW is the welded specimen of medium section, and EW is the welded specimen of end section. The other specimens are the threaded connection specimen of the inner lining tube. The first letter M or E is the connection position located in the middle section or the end section respectively, the Arabic numeral 6 or 9 refer the thread depth of 0.6 mm or 0.9 mm respectively, and the third letter A, B or C refers the thread length of 16.5 mm, 33 mm or 66 mm respectively.

**Table 2 materials-15-08619-t002:** Indices of steel tube.

Type	*f*_y_/MPa	*f*_u_/MPa	*E*_s_/GPa	*v* _s_	*δ*/%
Steel tube	420	570	215	0.28	20.7
inner lining tube	419	569	210	0.27	25.9

**Table 3 materials-15-08619-t003:** Mix proportion of concrete (kg/m^3^).

Cement	Water	Sand	Low-Grade Magnesite
432	168	558	1242

**Table 4 materials-15-08619-t004:** Comparison between calculated value and test value of bearing capacity of specimens.

No.	Test Value *N*_ue_/kN	Reference [1] *N*_uc_/*N*_ue_	Reference [24] *N*_uc_/*N*_ue_	Reference [25] *N*_uc_/*N*_ue_	Reference [26] *N*_uc_/*N*_ue_
M6A	1841	1.00	0.75	0.90	0.90
M6B	1764	1.05	0.78	0.94	0.94
M6C	1810	1.02	0.76	0.91	0.91
M9A	1539	1.20	0.90	1.07	1.07
M9B	1695	1.09	0.81	0.98	0.97
M9C	1729	1.07	0.80	0.96	0.95
E6A	1782	1.03	0.77	0.93	0.93
E6B	1914	0.96	0.72	0.86	0.86
E6C	1849	1.00	0.75	0.89	0.89
E9A	1857	0.99	0.74	0.89	0.89
E9B	1868	0.99	0.74	0.88	0.88
E9C	2130	0.87	0.65	0.78	0.77
Average value		1.02	0.76	0.92	0.91
Mean square deviation		0.079	0.059	0.071	0.071

## Data Availability

Not applicable.

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
