# Peer review of "Test on Compressive Performance of Concrete Filled Circular Steel Tube Connected by Thread through Inner Lining Tube"

_materials, 2022, doi:10.3390/ma15238619_

Round 1

Reviewer 1 Report

Dear Authors,

thank you very much for submitting this article.

Research on improving joints in concrete-filled steel pipes. 

As you state in your introduction, such a solution has many uses, but also its problems. You correctly state that one of them is the connection of larger pipes. In the introduction you list a large number of already defined and clearly tested research and solutions and then state what will be in the article, but I do not see a clear justification of this topic. 

The description of the selection of variants studied and the number of elements is insufficient. 

The description of the experiment and the evaluation of the results and the failur mode is fine.

I don't understand the meaning of mean and standard deviation for load-bearing capacigy 

I recommend adding a numerical model and a parametric study. 

The conclusions are totally inadequate. 

Author Response

Q1:The description of the selection of variants studied and the number of elements is insufficient.

A1:

Thanks for your suggestion. For the thread length, no references in this field were found during the parameter selection process, only those in the mechanical field[1-2]. As an exploratory test, we tried according to those references in the mechanical field, and found that the thread length is relatively short, and the connection effect is poor. When the thread length is long, it is only equivalent to improving the steel ratio of the specimen, which has no effect on the connection effect of the specimen. Therefore, the thread length Ds/2, Ds/4 and Ds/8 is selected in this paper according to the references and pre-experiment.

As for the thread depth, it should not be too small. It was found during the test that when the thread depth is lower than 0.6mm, it is easy to trip off. In addition, its depth cannot be too large. According to the Chinese standard [3-4], the strength of the specimen connected by welding is more than 80% of the strength of the unconnected specimen. Therefore, considering this factor, the relative value of the thread depth should be less than 20% of the steel tube wall thickness, otherwise the strength will not meet the standard. Therefore, the thread thickness is selected as 0.1 ts and 0.15 ts.

As the smaller the wall thickness of the steel tube is, the smaller the selection of the thread depth is. However, if the wall thickness is too large, the steel ratio is not unreasonable. At the same time, considering the conditions of the test loading equipment and the safety reserve, the corresponding steel and concrete strengths are selected, and the final cross-sectional dimensions is determined.

In practical engineering, the middle section is the weak section of the specimens, while the end section is the safe section of the specimens. But as mentioned earlier, this is an exploratory test. There are few references at present, so this paper designs inner lining tube in the weak section and the safe section respectively. If the effect of inner lining tube design in the weak section is not bad, it means that the inner lining tube can be applied to all sections. If only the extension of the concrete filled steel tube in safety section is studied, there may be problems in the actual project. That is, if the extension position of the steel tube is not at the column end, the steel tube can only be cut to change the connection position to the safety section, resulting in material waste. Subsequent tests show that the mechanical properties of the specimen connected by thread through inner lining in the middle section tube are still good.

[1] GB/T 197-2018 General Purpose Screw Thread Tolerances. Beijing: China Architecture Press, 2018

[2] GB/T 898-1988 Double end stud bm=1d. Beijing: China Standards Press, 1988

[3] GB 50661-2011 Code for welding of steel structures. Beijing: China Architecture & Building Press, 2011.

[4] GB 50017-2017 Standard for design of steel structures [S]. Beijing: China Architecture & Building Press, 2017.

The design philosophy of the test specimens is described in Section 1.1, and relevant references have been added. Please see the highlight text (in green colour) in Section 1.1.

Q2: I don't understand the meaning of mean and standard deviation for load-bearing capacigy

A2:Mean standard deviation is the positive square root of the arithmetic mean of the square of the deviation between the flag values of all units and their arithmetic mean. It is also a means to verify the rationality of formulas.

Q3:I recommend adding a numerical model and a parametric study. 

A3:This is a very good suggestion. We will continue to study the finite element part of it in the later stage, but the time for modification given by the editor is limited, so we will carry out finite element research after a large number of tests in the later stage

Q4:The conclusions are totally inadequate.
A4:The conclusion is supplemented. Please see the highlight text (in red colour) in Conclusion.

Reviewer 2 Report

Test on compressive performance of concrete filled circular steel tube connected by thread through inner lining tube

In this research, the authors proposed six inner lining tubes and threaded connections for lengthening the steel tube of concrete-filled circular steel columns. They tested twelve specimens with different connection positions, thread depth, and the thread length. The connected CFST columns were tested under axial load and then compared with specimens with welded connections and ordinary specimens.  The failure mode, load-strain curves, and compressive bearing capacity were recorded.

Please consider the following comments:

-          Could you please explain the reason for using low-grade magnesite concrete in the tested concrete-filled steel tubular columns.

-          The authors depended on comparing the proposed technique with the ordinary and the traditionally welded CFST columns with respect to the ultimate bearing capacity, and strain in the outer steel tube. However, the comparison should include as well the cost and the applicability of this technique in practical life.

-          In specimens with thread through inner lining with the letter (C), the thread length was very big (as a connection) (the connection is about one-third of the column height). The results of this column cannot be compared with the traditional welded column.   

-          More discussion and explanation for the results should be provided.    

-          Table 1 (Page 3): in the fourth column entitled (Weld/Thread position) The position of welding in the second and third rows for specimens MW and EW was not written.

-          Table 2 (page 4): please revise the title of the sixth column.

-          Please check the titles of the 3rd, 4th, 5th, and 6th columns in Table 4. (Reference source errors).

-          Please note that the arrows and names of the parts in many figures throughout the manuscript are not positioned on the figures correctly (Page1 - Figure1, Page2- Figure2,  Page4- Figure3,  Page5- Figure5)

Author Response

1. Could you please explain the reason for using low-grade magnesite concrete in the tested concrete-filled steel tubular columns.

Magnesite is a magnesium carbonate mineral (Mg CO3), which is also the main source of magnesium resources. In areas rich in magnesite resources, high-grade magnesium ore was preferentially selected in the process of development and utilization, resulting large number of low-grade magnesite and magnesite tail ore are idle, causing a great waste of resources and serious environmental pollution. How to deal with and utilize low-grade magnesite is the general problem to be solved at present. To solve environmental problems, this paper attempts to use low-grade magnesite mining as the coarse aggregate of concrete to make low-grade magnesite concrete filled circular steel tube.

2. The authors depended on comparing the proposed technique with the ordinary and the traditionally welded CFST columns with respect to the ultimate bearing capacity, and strain in the outer steel tube. However, the comparison should include as well the cost and the applicability of this technique in practical life.

The authors agree with the reviewer’s comments in this point. As we mentioned earlier in the article, the current connection methods mainly include welding, flange connection and grouting sleeve connection, which have certain limitations at this stage. So it is necessary to provide a new connection mode with high degree of standardization, convenient construction, reasonable mechanical properties and not susceptible to environmental and anthropic factors. The author thinks that the main contribution of this paper is to provide a new connection method and further realize the diversity of concrete filled steel tube extension. We have revised introduction, please see the highlight text (in green colour) in introduction.

3. In specimens with thread through inner lining with the letter (C), the thread length was very big (as a connection) (the connection is about one-third of the column height). The results of this column cannot be compared with the traditional welded column.

Specimen C is ordinary CFST without thread. It is only used to compare the difference between the unconnected specimen and the connected specimen.

4. More discussion and explanation for the results should be provided.

Thanks for your suggestion. In the 4 section, we added the discussion of results due to different parameter changes.

5. Table 1 (Page 3): in the fourth column entitled (Weld/Thread position) The position of welding in the second and third rows for specimens MW and EW was not written.

Thanks for your suggestion. This problem is revised in table 1.

6. Table 2 (page 4): please revise the title of the sixth column.

This problem is revised in table 2.

7. Please check the titles of the 3rd, 4th, 5th, and 6th columns in Table 4. (Reference source errors).

We compared the experimental results with reference 1 and references 18~19 respectively, so we found no new problems after checking.

8. Please note that the arrows and names of the parts in many figures throughout the manuscript are not positioned on the figures correctly (Page1 - Figure1, Page2- Figure2,  Page4- Figure3,  Page5- Figure5)

Thanks for your suggestion. We checked the positions of all arrows. They are correct in the manuscript. If there are still problems, we will contact the editor.

Reviewer 3 Report

The work is practical. The authors prepared a series of pipe samples connected with an internal threaded element (different lengths). The pipes were filled with concrete, and then the samples were compressed and analyzed for damage. Theconclusions are of a practical nature.

From the scientific point of view, the work does not bring anything new, but I believe that the practical nature of the results is worth publishing.

The presentation style and format of the article are okey. The figures are appropriate and reflect the content of the article. I don't feel qualified to judge about the English language.

Author Response

Thanks for your review.

Reviewer 4 Report

-          The title should be improved, making it more innovative.

-          Why are you conducting this study?

-          What is the objective of this study.?

-          What is the development trends of this research topic?

-          What is the code/s the you are following in the design processes?

-          Correct the reference errors in Table 4?

-          Rename section “. Analysis of test results” into “Results and discussion”.

-          What is the limitation of this study?

-          What is the potential application of this product in the construction industries?

-          The format of the list of references is not in line with the “guideline of author, please correct it?

-          There are several grammatical errors and typos, please correct them.

-          There are some sections need in-depth discussions, such as sections 3, 4, 5,?  

-          The conclusions come from the results, but it is still a simple summary of the values obtained. The academic evaluation is lacking. The following structure of the Conclusion is recommended: Research object, Research method, Brief description of the research process, Numbered short statements of new scientific results obtained by the author.

Author Response

1. Why are you conducting this study?

The current connection methods mainly include welding, flange connection and grouting sleeve connection, which have certain limitations at this stage. So it is necessary to provide a new connection mode with high degree of standardization, convenient construction, reasonable mechanical properties and not susceptible to environmental and anthropic factors.

2. What is the objective of this study.?

As we mentioned earlier in the article, to provide a new connection mode with high degree of standardization, convenient construction, reasonable mechanical properties and not susceptible to environmental and anthropic factors.

3. What is the development trends of this research topic?

As we mentioned earlier in the article, the current connection methods have many references. But the new connection mode basically no reference. The author thinks that the main contribution of this paper is to provide a new connection method and further realize the diversity of concrete filled steel tube extension.

4. What is the code/s the you are following in the design processes?

[1] GB/T 197-2018 General Purpose Screw Thread Tolerances. Beijing: China Architecture Press, 2018

[2] GB/T 898-1988 Double end stud bm=1d. Beijing: China Standards Press, 1988

[3] GB 50661-2011 Code for welding of steel structures. Beijing: China Architecture & Building Press, 2011.

[4] GB 50017-2017 Standard for design of steel structures [S]. Beijing: China Architecture & Building Press, 2017.

The design philosophy of the test specimens is described in Section 1.1, and relevant references have been added. Please see the highlight text (in green colour) in Section 1.1.

5. Correct the reference errors in Table 4?

We compared the experimental results with reference 1 and references 22~24(Original18~19) respectively, so we found no new problems after checking.

6. Rename section “. Analysis of test results” into “Results and discussion”.

Thanks for your suggestion. This problem is revised.

7. What is the limitation of this study?

The influence of slenderness ratio should be considered in the actual project of concrete-filled steel tube. Therefore, the research on the connection of short columns has no practical significance, but as we mentioned in the paper, for the threaded concrete-filled steel tube, axial tension and torsion is most unfavorable stress forms, followed by tension-bending, bending, compression-bending. Axial compression and shear are favorable stress forms for threaded concrete-filled steel tube, and the axial compressive performance is the most important and the most basic performance of concrete-filled steel tube. Therefore, we will study the performance of long columns and other stress modes in the subsequent research

8.What is the potential application of this product in the construction industries?

Providing a new connection mode with high degree of standardization, convenient construction, reasonable mechanical properties and not susceptible to environmental and anthropic factors, and Suggestions for calculating the axial compressive bearing capacity of low-grade magnesite concrete filled circular steel tube connected by thread through inner lining tube are suggested in construction industries.

9.The format of the list of references is not in line with the “guideline of author, please correct it?

We checked the form of references and made some adjustments in consultation with the editor.

10.There are several grammatical errors and typos, please correct them.

Thanks for your suggestion. Several grammatical errors and typos are revised.

11.There are some sections need in-depth discussions, such as sections 3, 4, 5,?

We added some discussions in 3 and 5 sections.

12.The conclusions come from the results, but it is still a simple summary of the values obtained. The academic evaluation is lacking. The following structure of the Conclusion is recommended: Research object, Research method, Brief description of the research process, Numbered short statements of new scientific results obtained by the author.

Thanks for your suggestion. Conclusion section is revised. Please see the highlight text (in green colour) in Conclusion.

Round 2

Reviewer 1 Report

After studying your answers in detail, and after studying the modifications at my suggestion, and at the suggestion of the other reviewers, I must conclude that the paper is of a better standard.

After a second reading, I recommend including in the introduction a mention of the possibilities of numerical models that you can use and apply in further research on your design.

For example:

10.1007/s13296-019-00210-w 

10.3390/ma14216573

Author Response

We have added relevant references in the abstract, please see the highlighted yellow part.

Reviewer 4 Report

I have no more comments to provide.

Author Response

Thank you for your review